# Carotenoid composition and sequestration in cassava (*Manihot esculentum* Crantz) roots

**Margit Drapal**[1], **Tatiana M. Ovalle Rivera**[2], **Luis Augusto Becerra Lopez-Lavalle**[2], **Paul D. Fraser**[1]*

1 School of Biological Sciences, Royal Holloway University of London, Egham, United Kingdom,
2 International Center for Tropical Agriculture (CIAT), Cali, Colombia

* P.Fraser@rhul.ac.uk

**Data Availability Statement:** All relevant data are within the article and its Supporting Information files. Additional raw data can be access through

## Abstract

Cassava (*Manihot esculentum Crantz*) is a staple food source for many developing countries. Its edible roots are high in starch but lack micronutrients such as β-carotene. In the present study, analysis of pedigree breeding populations has led to the identification of cassava accessions with enhanced β-carotene contents up to 40 µg/g DW. This represents 0.2% of the Recommended Daily Allowance (RDA) for vitamin A. The β-branch of the carotenoid pathway predominates in cassava roots, with dominant levels of β-carotene followed by other minor epoxides of β-ring derived carotenoids. Metabolomic analysis revealed that steady state levels of intermediary metabolism were not altered by the formation of carotenoids, similar to starch and carbohydrate levels. Apocarotenoids appeared to be independent of their carotenoid precursors. Lipidomic analysis provided evidence of a significant positive correlation between carotenoid and lipid content, in particular plastid specific galactolipids. Proteomic analysis of isolated amyloplasts identified the majority of proteins associated with translation and carbohydrate/starch biosynthesis (e.g. glucose-1-phosphate adenylyltransferase). No carotenoid related proteins were detected even in the highest carotenoid containing lines. Carotenoids were associated with fractions typically annotated as plastoglobuli and plastid membranes (particularly the envelope). Proteomic analysis confirmed these structures apart from plastoglobuli, thus potentially amyloplast structures may not contain classical plastoglobuli structures.

## Introduction

Cassava plants (*Manihot esculentum* Crantz) have been cultivated in Central and South America for ~9000 years and have shown a high potential to adapt to different environments [1–3]. Studies have shown feasible yields of cassava roots on marginal soils, and under drought and other biotic and abiotic stresses [1, 3]. Hence, cassava is now grown in low- and middle-income countries throughout the tropics and subtropics and presents the second most important source of calories in Africa [4–7]. However, the micronutrient and carotenoid (<1 µg/100 g β-carotene) content of cassava roots is comparatively low [8]. In order to improve the nutritional quality of cassava roots, natural variation has been evaluated.

Mendeley Data (Drapal and Fraser, 2023): https://data.mendeley.com/datasets/dd6whgr49c/1.

**Funding:** The authors received the following funding, awarded to Paul Fraser: Consortium of International Agricultural Research Centers, Research on Roots, Tubers and Bananas; Biotechnology and Biological Sciences Research Council (grant number BB/T008946/1); and FP7 Food, Agriculture and Fisheries, Biotechnology (grant number 613513).

**Competing interests:** The authors have declared that no competing interests exist.

**Abbreviations:** CIAT, International Center for Tropical Agriculture; DAG, diacylglycerides; DGDG, digalactosyldiacylglycerol; DW, dry weight; GC-MS, gas chromatography mass spectrometry; HexCer, hexoside ceramide; LC-MS, liquid chromatography mass spectrometry; MAG, monoacylglycerides; MGDG, Monogalactosyldiacylglycerol; MPS, multipurpose sampler; NIRS, near infra-red spectroscopy; OR, orange; PA, phosphatidic acid; PC, phosphatidyl choline; PCA, principal component analysis; PCs, principal components; PE, phosphatidyl ethanolamine; PG, phosphatidyl glycerol; PI, phosphatidyl inositol; PSY, phytoene synthase; pVA, precursor for Vitamin A; RDA, Recommended Daily Allowance; SDS-PAGE, Sodium dodecyl-sulphate polyacrylamide gel electrophoresis; SPME, solid phase microextraction; SQDG, sulfoquinovosyldiacylglycerol; Susy, sucrose synthase; TAG, triacyl glycerol; TCA, tricarboxylic acid; UPLC-DAD, ultra-pressure liquid chromatography–diode array detector; VAD, Vitamin A deficiency.

Screening of cassava landraces observed some genotypes e.g. BRA1A and COL638 naturally contain β-carotene (20–30 μg/g dry weight (DW)) in the root tissue [9, 10]. β-Carotene is the most efficient precursor for Vitamin A (pVA), which is an essential vitamin in the human diet, required for healthy vision and cell differentiation [11, 12]. Vitamin A deficiency (VAD) is a result of a low pVA diet and several complementation strategies (e.g. diverse diet, food fortification and supplementation) have been developed to address and overcome VAD [13]. Biofortification of cassava has been a recent approach implemented by programmes such as HarvestPlus (www.harvestplus.org) through conventional breeding, which utilised the genetic variability already present in landraces [14–16]. More recently, genetic transformation with bacterial carotenoid genes was utilised to increase the levels of β-carotene to 60–75 μg/g DW [17, 18]. Genetic assessment of selected genotype panels enabled the elucidation of underlying genes involved in high β-carotene content [19, 20]. These studies have established key enzymes e.g. PHYTOENE SYNTHASE (PSY) and the ORANGE (OR) chaperone protein for carotenoid accumulation. However, the characterisation of amyloplasts, the site of starch and carotenoid biosynthesis and sequestration in cassava roots, has not been carried out.

Studies in *Solanum tuberosum* (potato) and *Ipomoea batatas* (L.) Lam (sweet potato) have highlighted significant remodelling of subplastidal components and adaptation of metabolic pathways to accommodate carotenoid accumulation [21, 22].

As part of the breeding strategy for increased carotenoid content in cassava roots, genetic screening was combined with near infra-red spectroscopy (NIRS) and liquid chromatography (LC) spectroscopy, facilitating the selection of outstanding off-spring or potential parents [15, 23]. NIRS is the preferred tool due to its amenability and ease of use [23, 24]. However, other metabolomics techniques combining gas or liquid chromatography (GC and LC) with photo diode array (PDA) or mass spectrometry (MS) detectors can provide quantitative levels of β-carotene, its precursors (e.g. phytoene, TCA cycle, glycolysis) and products (e.g. apocarotenoids) and metabolites involved in sequestration processes (e.g. membrane lipids) [25, 26]. Additionally, untargeted analysis of polar extracts, comprising the majority of components of the central metabolism (glycolysis, TCA cycle, amino acids), may contribute to elucidate whether respective phenotypes in a breeding population affect the cellular metabolism or specific metabolic pathways [27–29].

A combination of metabolomics techniques was applied to three carotenoid breeding populations derived from cassava landraces (S1 Table). The populations were harvested in 2015 to measure carotenoid content and the composition of polar extracts. One population (GM3736/3732) was regrown in 2017 and 2020, with a varied number of lines due to the performance of the different lines in the field. The carotenoid content was measured each harvest year and showed different results for the same lines, indicating an environmental influence on the carotenoid biosynthesis of cassava roots. Based on the results of GM3736/3732 in 2015, eight lines representing low, medium and high carotenoid content, were grown specifically in 2017 and 2020 for a more detailed study of their biochemical and subcellular composition. The biochemical analysis involved measurements of starch, protein and total lipid content. The subcellular analysis involved extraction and fractionation of amyloplasts, the site of carotenoid biosynthesis and sequestration. Proteomics and lipidomics were applied to characterise the compartments of amyloplasts and the data correlated to β-carotene content. Results highlighted a complex regulation of the carotenoid metabolism independent of central metabolism. However, the biochemical and subcellular fractionation data showed a direct correlation of β-carotene to lipid composition of the amyloplast. Hence, the analysis of GM3736/3732 harvested in 2020 included lipid analysis of the whole root tissue.

## Materials and methods

### Plant material

CIAT's standard field conditions were used to cultivate all cassava genotypes. Three different cassava population (GM3732/36, GM5270 and GM5309) bred for increased β-carotene content were used (S1 Table). Cassava roots were harvested ten months after planting, and one root per plant immediately frozen in liquid nitrogen and lyophilised for metabolomics analysis. The lyophilised tissue was stored at -20˚C and ground to a fine powder shortly before analysis, which was kept at -20˚C. For subcellular fractionation, three cassava roots were harvested, immediately dipped in liquid paraffin to cover the whole root [30], shipped to the UK and processed upon arrival at Royal Holloway University of London. The processing was performed separately for each replicate and included peeling the roots and cutting them into chunks (~2 cm width x ~7 cm length) before preparation for plastid extraction and subplastidal fractionation. A small portion of the peeled roots were frozen in liquid nitrogen, lyophilised and stored at -20˚C for metabolomics analysis.

### Metabolomics analysis

**Carotenoid analysis.** Lyophilised tissue of each variety was weighed in triplicate (50 mg each) and extracted with chloroform/methanol on ice as previously described [31]. After phase separation, the whole chloroform extract containing carotenoids was dried down through centrifugal evaporation. Immediately before analysis, the dried aliquots were resuspended in ethyl acetate (50 μL) and any particles were separated through centrifugation for 5 min at full speed. An aliquot of the supernatant (45 μL) was transferred to a glass insert and analysed with a Acquity™ UPLC-PDA (Waters, UK) with a BEH column (2.1x100 mm, C18, 1.7 μm). The injection volume was 5 μL and quantification was performed as previously published [32, 33] (S2 Table).

**Analysis of polar metabolites.** Lyophilised tissue of each variety was weighed in triplicate (10 mg each) and extracted with chloroform/methanol/water and analysed with 7890A GC system coupled to a 5795C MS detector (Agilent Technologies, Inc.) in splitless mode with a gradient of 8˚C/min, modified from a previously published method [34]. For derivatisation, an aliquot of the polar extract (120 μL) was dried down and internal standard $d_4$-succinic acid (10 μg) added. Data analysis and compound identification was performed with AMIDS (v2.71) (S3 Table).

For untargeted analysis by LC-MS of cassava populations GM5270 and GM5309, an aliquot of the GC-MS extraction (700 μL) was dried down and resuspended in methanol/water (1:1, 200 μL) after the addition of internal standard genistein (1.25 μg). Chromatographic analysis was performed with an UHPLC UltiMate 3000 (Dionex Softron) linked to a MAXIS UHR-Q-TOF (Bruker Daltonics) [35]. The samples (5 μl injection volume) were separated with an UltraHTPro C18 column (100x2 mm, i.d. 2 μm, YMC, UK). Data was extracted as previously described with R package metaMS [36] (S4 and S5 Tables).

For LC-MS analysis of cassava population GM3736/3732, an aliquot of the GC-MS extraction (100 μL) and internal standard genistein (1 μg) were transferred to a glass insert and analysed with a 1290 Infinity II coupled to a 6560 Ion Mobility Q-TOF (Agilent Technologies, Inc.) as previously published [37]. The sample (5 μl injection volume) was separated on a ZORBAX Eclipse Plus C18 (2.1x50 mm, 1.8 μm, Agilent Technologies, Inc.). Data analysis was performed with Agilent Profinder (v10.0 SP1, Agilent Technologies, Inc.). The settings for peak selection included a peak height >1000 counts, mass features detected between 0.5–10 min, a retention time tolerance of 0.2 min and a mass tolerance 20 ppm (S6 Table).

All samples were analysed as three technical replicate and average values used for statistical analysis and data processing.

**Apocarotenoid and volatile analysis.** Lyophilised powder (300 mg) was weighed into a brown glass vial with screw cap top and $d_8$-acetophenone (21.1 µg) was added. The samples were analysed with solid phase microextraction (SPME) GC-MS (Agilent 7890 B and 5977 B MSD, Agilent, Inc.) with a Gerstel multipurpose sampler (MPS) (Gerstel, Germany), as previously published [38]. The sample was incubated at 60˚C and 300 rpm for 25 min and the metabolites extracted with a divinylbenzene, carboxen, polydimethylsiloxane (DVB/CAR/PDMS) fibre (50/30 µm, Supelco, USA) with 23 gauge needle for 30 min at 300 rpm. Samples were analysed as three technical replicates. Data analysis and compound identification was performed with AMDIS (v2.71) (S7 Table).

**Analysis of glyceride lipids.** Lyophilised powder (10 mg) was extracted with chloroform/methanol (1 mL, 2:1) for 20 min and 40 rpm at 4˚C. Samples were centrifuged at full speed and an aliquot (500 µL) dried down. The samples were resuspended in ethanol (500 µL) and analysed (2 µl injection volume) with a Nucleodur C8 gravity column (4.6x50 mm, 1.8 µm, Macherel-Nagel Hichrom) on a Infinity II 1290 UHPLC coupled to a 6550 iFunnel QTof (Agilent Technologies, Inc.) as previously published [21]. Data analysis was performed with Agilent Profinder (v10.0 SP1, Agilent Technologies, Inc.) and quantified with an authentic standard mono, di and triglyceride mix (Supleco, USA) (S8 Table).

## Biochemical assessment for starch, protein and lipid content

Lyophilised powder from cassava roots for subcellular fractionation was weighed out in triplicate for starch (50 mg), protein (11 mg) and lipid (2 g) assessment as previously described [21]. Soluble starch (Sigma, UK) and bovine serum albumin (Sigma, UK) were used for calibration curves, whereas lipid content was weighed directly.

## Subplastidal fractionation

Eight cassava genotypes (GM3736-2, GM3736-29, GM3736-44, GM3736-50, GM3736-71, GM3736-78, GM905-57 and GM905-60) were selected for subplastidal fractionation and analysed at two harvest years, 2017 and 2020. Plastids were extracted from peeled cassava roots (~100 g), lysed and separated on a sucrose gradient as previously described for potato [22]. The sucrose gradient was stored as aliquots (1 mL), referred to as "fractions", which were collected from the top (1) to the bottom (34). Fractions 1–31 of GM3736-71 and GM3736-78 and fractions 1, 2 and 20–23 for the other six genotypes were extracted with methanol (200 µL) and chloroform (600 µL) for 20 min at 4˚C. After centrifugation at 4˚C and full speed for 5 min, the chloroform phase representing the lipid extract was dried down and resuspend in chloroform (100 µL). The phosphor- and galactosyl lipids in the lipid extracts were analysed in MS/MS mode with Infinity II 1290 UHPLC coupled to a 6550 iFunnel QTof (Agilent Technologies, Inc.), as previously described [21]. The samples (10 µL injection volume) were separated with an Atlantis HILIC Silica column (2.1x100 mm, 3 µm, Waters, UK). Data analysis was performed with Agilent Profinder (v10.0 SP1, Agilent Technologies, Inc.) and quantified with an authentic standard of monogalactosyldiacylglycerol (MGDG) (Avanti Polar Lipids, Inc.) (S9 Table).

After the lipid extraction, proteins in fractions 1, 2, 20–25 and 29 of GM905-60, GM3736-71, GM3736-78 were precipitated through addition of ice-cold (-20˚C) methanol (600 µL) at 4˚C for 10 min, followed by centrifugation for 5 min at full speed and 4˚C and removal of the supernatant. The protocol published by Nogueira et al. (2013) was used with minor modifications. Protein pellets were denaturised in sodium dodecyl sulphate (SDS) buffer (fractions 1

and 2 in 20 µL; fractions 20–25 in 50 µL; fraction 29 in 75 µL) and separated by SDS-polyacryl-amide gel electrophoresis (PAGE). The protein gels were separated into three sections: 0–20 kDa, 20–30 kDa and >30 kDa, before trypsin digestion of the individual fraction lanes. Digested peptides were analysed with an AdvanceBio Peptide Map column (2.1x100 mm, 2.7 µm, Agilent Technologies, Inc.) and an Infinity II 1290 UHPLC coupled to a 6550 iFunnel QTof (Agilent Technologies, Inc.). Data analysis was performed with Mascot Distiller (v2.4.2.0, Matrix Science) and confirmed with Spectrum Mill MS Proteomics Workbench (Rev B.06.00.201, Agilent Technologies, Inc.). Search parameters included modifications carboxy-methylation and oxidation of methionine, as previously published [31]. UniProt database for *Manihot esculentum* (Taxon ID 3983), downloaded in August 2021, was used to identify digested peptides (S11 Table).

## Data analysis

Processed data was subjected to statistical analysis and graphically displayed with Simca® 17 (Sartorius), XLSTAT (2017, Addinsoft) and GraphPad Prism (9.4.1, GraphPad Software, California USA). Statistical analysis included parametric ANOVA, simple linear regression and Pearson correlation with *P*-value ($<0.05$).

All data is available with the manuscript. Additional raw data can be access through Mendeley Data [39]: https://data.mendeley.com/datasets/dd6whgr49c/1.

## Results

### Carotenoid composition of cassava roots

Carotenoid analysis was performed for cassava roots with eight different root phenotypes, white to intense yellow, selected in 2015. The phenotypes were represented by the following varieties GM3736-2, GM3736-29, GM3736-44, GM3736-50, GM3736-71, GM3736-78, GM905-57 and GM905-60. The analysis identified 23 carotenoids (Fig 1) and showed that β-carotene was the carotenoid with the highest concentration in all root phenotypes. There was a

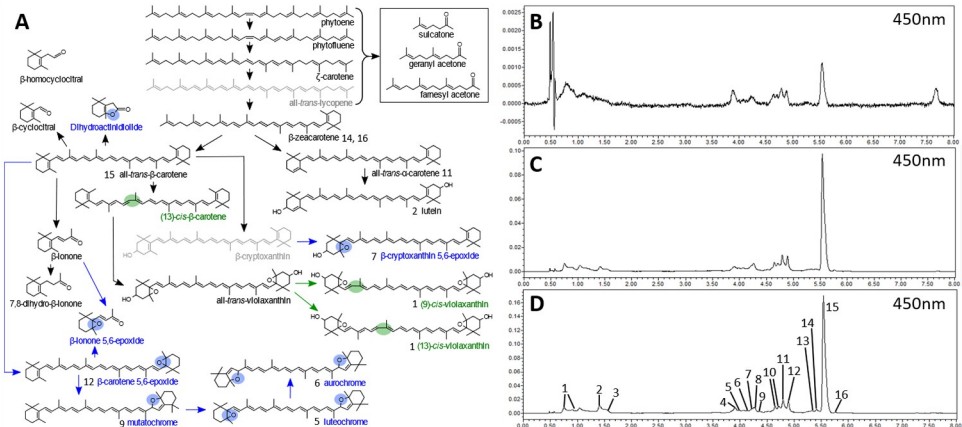

**Fig 1.** Carotenoid biosynthetic pathway (A) based carotenoid and apocarotenoid analysis in cassava roots (B-D). (A) Carotenoids containing epoxides or cis-isomers are labelled in blue and green, respectively. Carotenoids not detected in cassava roots are displayed grey. Representative chromatograms analysed by UPLC-DAD at 450nm are displayed for cassava roots containing 0.7 µg (B), 8.4 µg (C) and 14.4 µg (D). Peaks in the chromatograms were identified as cis-violaxanthin (1), all-trans-lutein (2), unknown (3, 4, 8, 10), luteochrome (5), aurochrome (6), β-cryptoxanthin 5,6-epoxide (7), mutatochrome (9), all-trans-α-carotene (11), β-carotene 5,6-epoxide (12), di-cis-ζ-carotene (13), β-zeacarotene isomers (14, 16), all-trans-β-carotene (15) and 13-cis-β-carotene (15).

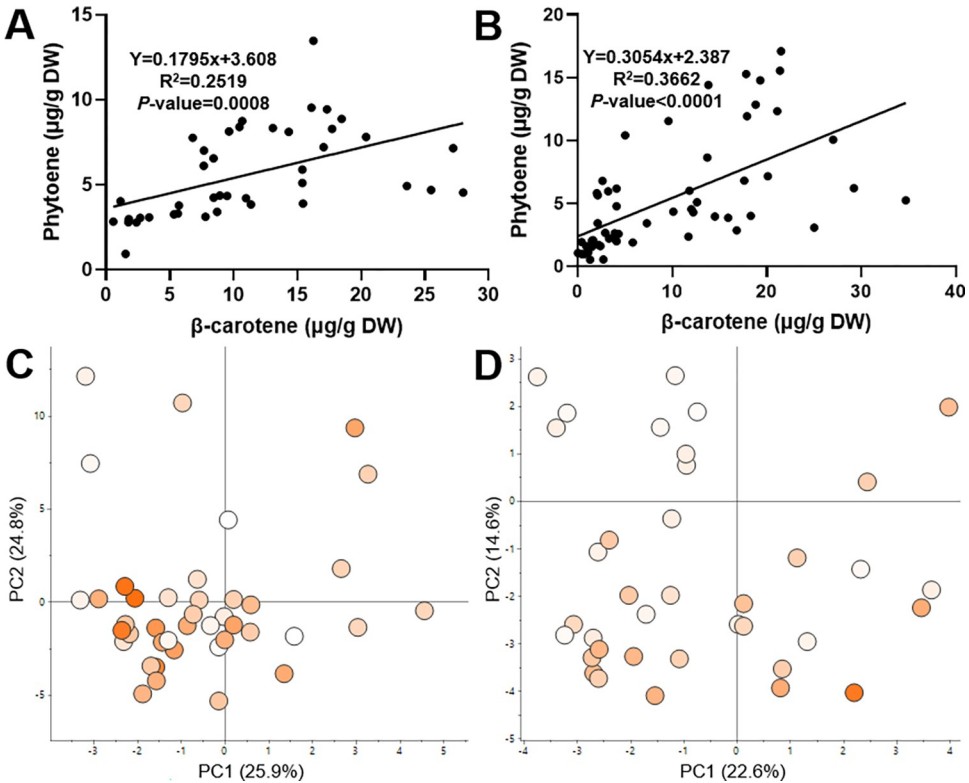

**Fig 2.** Correlation between phytoene and β-carotene (A, B) and volatile analysis (C, D) of a cassava carotenoid breeding population GM3732/3736 grown over two different years (2017 and 2020). Linear regression (line) was displayed for the correlation (A, B), as well as the equation, Chi-square goodness of fit ($R^2$) and significance (*P*-value). The β-carotene content of each variety is indicated by a colour range of white (lowest) to orange (highest). The colour range was scaled to the lowest and highest β-carotene of each crop: 0.6–28 μg/g DW (C) and 0–34.7 μg/g DW (D). The data present shows the average of three technical replicates.

direct correlation between β-carotene content and the intensity of the root colour phenotype. If β-carotene was present in the analysed cassava varieties, all other carotenoids could also be detected. This can be observed in representative chromatograms at 450nm of cassava roots with 0.7μg, 8.4μg and 14.4μg β-carotene per g DW (Fig 1). Correlation analysis between β-carotene and phytoene content (Fig 2A and 2B) indicated a significant positive correlation with a low goodness of fit ($R^2 < 0.4$).

Further analysis for volatile isoprenoids was performed on the whole cassava population GM3736/3732, from which the eight cassava varieties were selected (Fig 2C and 2D). The GM3736/3732 population was grown over two different years 2017 and 2020. In 2017, the GM3736/3732 population contained β-carotene levels from 0.6–28 μg/g DW and in 2020, 0–34.7 μg/g DW. The PCA score plots of volatile metabolites showed no grouping based on the β-carotene content of the individual varieties (Fig 2C and 2D).

The volatile analysis showed that β-cyclocitral, β-ionone, β-ionone-5,6-epoxide, dihydroactinidiolide, sulcatone and geranyl acetone were present in almost all samples. The apocarotenoids in the present study showed a strong positive correlation (~0.4–0.9) with each other and no statistically significant correlation to β-carotene and phytoene. The apocarotenoids analysed in the 2017 crop showed a negative correlation trend (-0.1 to -0.2) to β-carotene. In 2020, the correlation coefficients to β-carotene were between -0.1 and 0.1, with the exception of

β-ionone. This apocarotenoid had a significant negative correlation (~-0.3) to β-carotene and β-carotene-5,6-epoxide.

## Carotenoid sequestration in plastid derived fractions of cassava roots

The eight cassava varieties used for the in-depth carotenoid analysis (GM3736-2, GM3736-29, GM3736-44, GM3736-50, GM3736-71, GM3736-78, GM905-57 and GM905-60) were studied to elucidate the carotenoid sequestration in cassava amyloplasts. The peeled cassava roots were processed with established protocols and amyloplasts studied through proteomic and metabolic techniques identifying subcellular structures. This included a separation of subcellular components of the amyloplasts on a sucrose gradient (Fig 3A). The gradients highlighted no

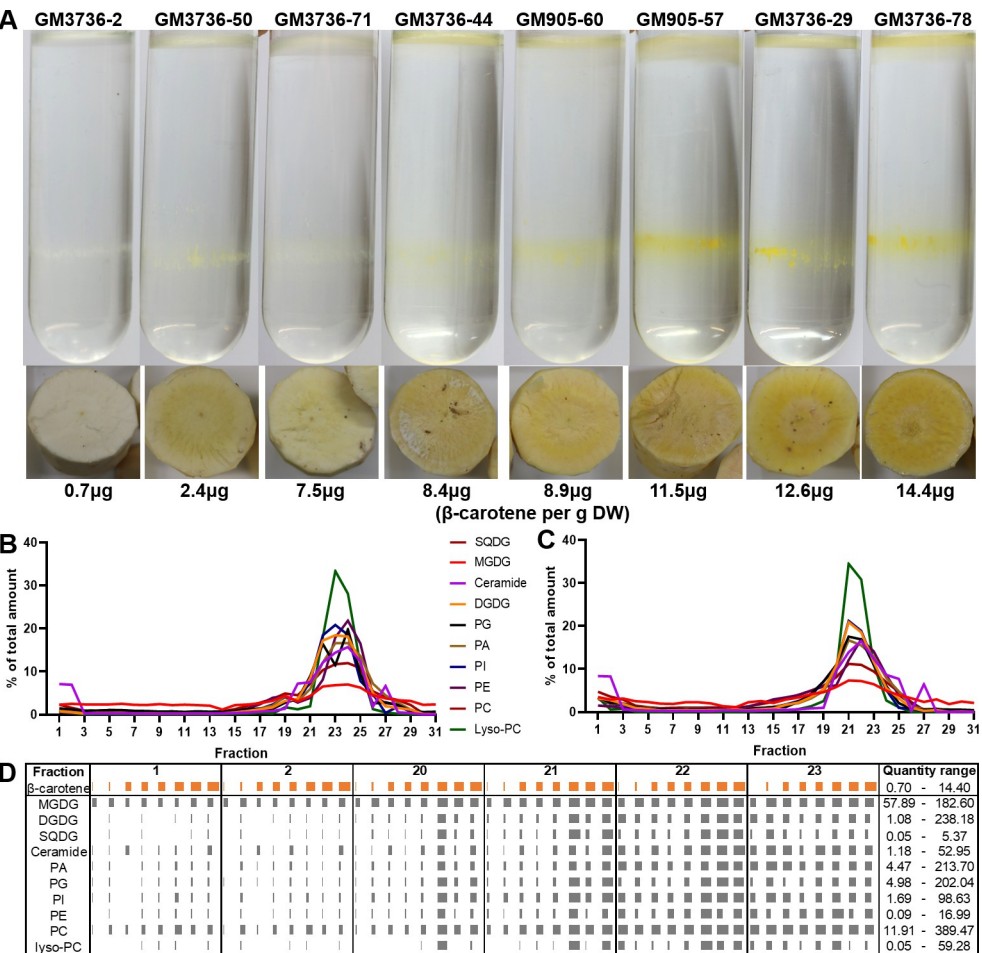

**Fig 3.** Subplastidal fractionation of amyloplasts from eight cassava root phenotypes harvested in 2017 (A), the percentual distribution of nine lipid classes throughout the gradient (B, C) and actual quantities of lipids in selected fractions of the eight cassava varieties. (A) The amyloplast components were separated with a sucrose gradient. Each gradient is display above a cross-section picture of the individual phenotypes and the respective β-carotene content. (B, C) Lipid profiling of sucrose gradient of GM3736-71 (B) and GM3736-78 (C). The content of each lipid class was calculated as the total sum of the five to six most prominent masses detected. The percentage of each lipid class for the individual fractions was displayed. (D) Quantification of ten lipid classes in selected fractions in eight cassava varieties. The content of β-carotene and the individual lipid classes was displayed as μg/g DW and ng/fraction, respectively. Each lipid class was calculated as the sum of the five to six most prominent lipids. The quantities were scaled between the lowest and highest quantity measured for each lipid individually and are listed on the right side under "Quantity range". The orange bars indicate the β-carotene content of the eight root phenotypes, as displayed in panel A.

**Table 1. Plasma membrane associated proteins identified in subcellular fractions of cassava amyloplasts.** The database code relates to Uniprot ID 3983 –*Manihot esculenta*. Fraction were the proteins were detected are listed. Brackets next to the fraction number indicate that the protein was found solely in GM3736-71 (L), GM905-60 (M) or GM3736-78 (H).

| Database code | Protein name | Molecular function | Fractions |
|---|---|---|---|
| A0A2C9WFF9 | 14-3-3 domain-containing protein | Protein localisation; signal transduction | 22–25 |
| Q1AP39 | 14-3-3 protein | Protein localisation | 22–25 |
| A0A1B4ZAZ5 | ATP synthase | Proton transmembrane transporter activity | 21 (H), 23, 24 (M), 25 |
| A0A2C9UPJ8 | FAS1 domain-containing protein | anchored component of plasma membrane | 23 (H), 24, (M, H), 25, 29 |
| A0A2C9UYF6 | Phospholipase D | Cleaves phosphodiester bond of PC and PE | 21 (H), 22 (L, M), 23, 24 (M, H), 25 (L) |
| V9M4W7 | Plastid 16-kDa outer membrane protein | Transmembrane transporter activity | 1 (M, H), 20–25 |
| A0A2C9UBP1 | Profilin | Actin monomer binding | 23, 24 (H), 25 |
| A0A2C9UL58 | Tubulin | GTPase activity; structural constituent of cytoskeleton | 23, 24 (H), 25 |
| A0A251JQ24 | Uncharacterized protein | Constituent of the cytoskeleton | 1, 21–25 |
| A0A251JTZ2 | Vacuolar proton pump | Proton transmembrane transporter activity | 23, 24 (H) |

difference of amyloplast component structures between the root colour phenotypes. The most prominent difference was the different colouration of the layer at the top of the gradient and the layer just below the middle of the gradient. The colour change of the two layers showed a visual correlation with the β-carotene content and appeared more yellow in varieties with higher β-carotene content.

For identification of the amyloplast components in the sucrose gradient, protein and lipid analysis was performed. For protein analysis, three varieties (GM3736-71, GM905-60 and GM3736-78) with different root colour phenotype were chosen and confirmed the same distribution of subplastidal components throughout the sucrose gradient. The proteins of fraction 1 and 2 (referred to as 1 from here on), 20–25 and 29 of GM3736-71, GM905-60 and GM3736-78 were digested and compared to the most up-to-date Uniprot protein database for *M. esculenta* to identify the detected peptides. For confirmation of amyloplast components, only proteins detected in all three varieties were selected (S11 Table). Identified proteins associated with plastid membrane were involved in transport (e.g. ATP synthase and ADP/ATP carrier protein), lipid metabolism (e.g. MlaD and FabA domain-containing proteins) and protein localisation and cytoskeleton regulation (14-3-3 protein and profilin) (Table 1). Plastid 16-kDa outer membrane protein, an integral part of the outer plastid membrane and part of the transmembrane transporter machine, was detected in fractions 20–25 and only in fraction 1 of GM905-60 and GM3736-78. An "uncharacterised protein", established as constituent of the cytoskeleton was identified in the same fractions as the latter protein. The number of proteins identified in fractions 23 and 25 indicate two layers of similar structure within the broad white to yellow band visible in the sucrose gradient.

A large proportion of the identified proteins were classified as ribosomal proteins and proteins associated with cytoplasm, based on the Uniprot database. Several proteins associated with carbohydrate metabolism were detected in fractions 20 to 29, indicating that the lower fractions of the gradient represented the stroma. These proteins included alpha-1,4 glucan phosphorylase, UTP glucose-1-phosphate uridylyltransferase and ribulose bisphosphate carboxylase large chain. Sucrose synthase was detected in fractions 20–25 and ribulose bisphosphate carboxylase small chain in fractions 23–25. Fructose bisphosphate aldolase, glucose-1-phosphate adenylyltransferase, glyceraldehyde-3-phosphate dehydrogenase, malate dehydrogenase were detected in fractions 23, 25 and 29. Only an uncharacterised protein with ribulose-biphosphate carboxylase activity and a Aamy domain-containing protein/1,4-alpha-glucan branching enzyme were also identified in fraction 1. These data suggests that i) the

subplastidal fractionation technique is not accurate enough to separate the membranal struc-
tures from the stroma fractions or ii) in cassava amyloplasts some stromal proteins are mem-
brane associated.

Lipid analysis was performed on all fractions of GM3736-71 and GM3736-78 (S1 Fig).
Eleven different classes of lipids were identified and were detected throughout the gradient.
The lipid classes included triacyl glycerols (TAG), MGDG, digalactosyldiacylglycerol
(DGDG), sulfoquinovosyldiacylglycerol (SQDG), ceramides, phosphatidic acids (PA), phos-
phatidyl glycerols (PG), phosphatidyl inositols (PI), phosphatidyl ethanolamine (PE), phos-
phatidyl cholines (PC) and lyso-PC. The analysis showed that TAG was overloaded in all
samples and was analysed separately with an optimised method for the whole cassava caroten-
oid population. The relative quantification of lipids in the samples showed that fractions 19–25
had the highest content of lipids and suggested that these fractions are membranal structures
(Fig 3B and 3C). Fraction 1 also showed higher percentage of lipids, compared to the following
ten fractions, which indicates the presence of a lipid related structure. Based on the lipid analy-
sis of GM3736-71 and GM3736-78, fractions 1–2 and 20–23 were analysed for the other six
cassava varieties. The chromatographic analysis of the fractions highlighted five to six domi-
nant molecular features for each of the lipid classes/chromatographic peaks, which showed a
significantly higher intensity and therefore higher peak area. These molecular features were
selected to calculate the quantity of the individual lipid classes. All detected lipids were
enriched in fractions 22 and 23. MGDG and PC were present in similar amounts in fraction 1
compared to the lower fractions 20–23. These two lipid classes showed a positive correlation to
β-carotene content, with the exception of fraction 23 (Fig 3C, S10 Table). This fraction seemed
to have similar levels of several lipids for all eight genotypes and showed no significant correla-
tion values. MGDG had high (>0.71) significant correlation values in fraction 20–22, indicat-
ing β-carotene levels could have an effect on plastid membrane composition. PC showed
similar results with correlation values >0.76 in fraction 1, 20 and 21. Fractions 20 showed the
most significant correlation values between lipid and β-carotene content, followed by fraction
21. The three cassava varieties with β-carotene content >10 µg/g DW had ~1.5- to >8-fold
higher lipid levels in fraction 20/21 compared to the other five cassava varieties with a β-caro-
tene content lower than 10 µg/g DW, supporting the hypothesis of a distinct plastid membrane
composition based on β-carotene content.

### Effect of increased β-carotene content on metabolism of cassava roots

The analysis of plastidal fractions indicated changes of the amyloplast in cassava varieties with
low and high β-carotene content (GM3736-2, GM3736-29, GM3736-44, GM3736-50,
GM3736-71, GM3736-78, GM905-57 and GM905-60). The analysis was performed for the
same eight cassava varieties harvested in 2017 and 2020 for subplastidal elucidation. The data
indicates some environmental influences on the cassava roots as the carotenoid levels were
~1.2- to ~1.9-fold higher in 2020. The lipid content showed no difference between the harvest
years, whereas starch was up to 50% lower in 2020 and protein levels increased by ~1.3- to
~2-fold (Fig 4B–4D). The proximate analysis showed no clear trend between starch, lipid and
protein content in relation to the β-carotene content (Fig 4). Hence, linear regression was used
to establish a significant correlation of lipids with a goodness of fit of ~0.59 for both harvest
years (Fig 4C). The lipid content for lines with medium β-carotene content (e.g. GM3736-71
and GM3736-71) was similar to that of the high β-carotene lines (e.g. GM3736-29), which
probably contributed to the medium correlation value. The starch content in 2020 showed
similar results with a significant correlation value (~0.49) and similar values between medium
and high lines (Fig 4B).

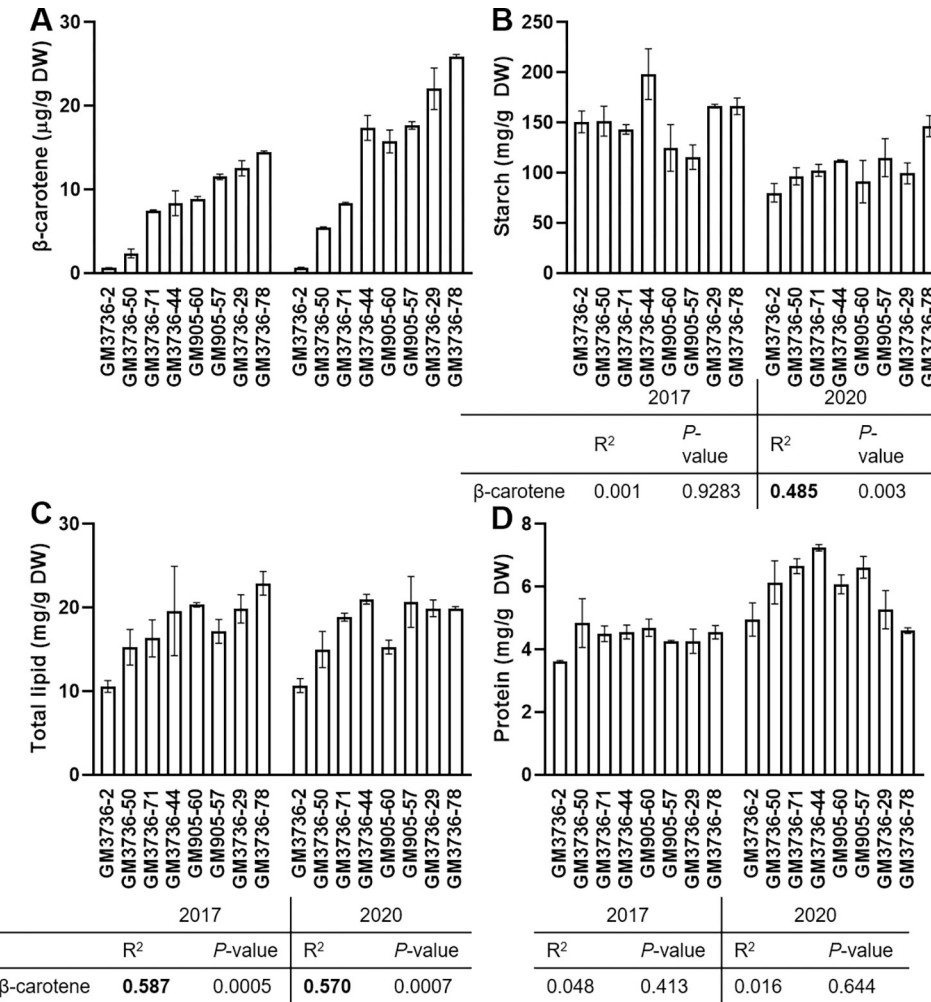

**Fig 4. Biochemical assessment of cassava roots with different β-carotene content.** The varieties were cultivated in 2017 and 2020 and are displayed separately on the left and right side of each bar chart, respectively. The assessments evaluated total content of (A) β-carotene, (B) starch, (C) lipids and (D) protein. Values are displayed as an average of three biological replicates with standard deviation. Correlation values of starch, lipid and protein to β-carotene content was displayed below the bar charts and included Chi squared goodness of fit ($R^2$) and significance (*P*-value).

A more in-depth analysis was performed on the carotenoid population, including analysis of glyceride lipids (Fig 5), GC-MS analysis of primary metabolites (Fig 6) and metabolite profiling by LC-MS analysis (Fig 7). The analysis of glyceride lipids was performed with extracts of freeze-dried ground roots of the cassava harvest in 2020 (Fig 5). The chromatograms showed a similar profile for all cassava varieties with seven distinct peaks. Comparison with a mono-, di- and triolein standard indicated that the first peak (1.5 min) comprised monoacylglycerides (MAG), the following three peaks (2.1–2.7 min) comprised diacylglycerides (DAG) and the remaining three peaks (4.4–4.8 min) comprised TAG. The fragmentation pattern of the peaks enabled the identification of five glyceride lipids: DAG(14:1/14:1), DAG(18:3/22:6), TAG(16:0/C18:1/C18:2), TAGC16:0/C18:2/C18:2) and TAG(C16:0/C18:1/C18:1). The total glyceride lipid and total TAG content was quantified and displayed as a violin plots, visualising the range and density of the lipid quantities in a two-dimensional manner (Fig 5A and 5B). The varieties of the cassava population were grouped based on their β-carotene content and showed no significantly different levels of total glyceride lipids or TAGs. Correlation analysis

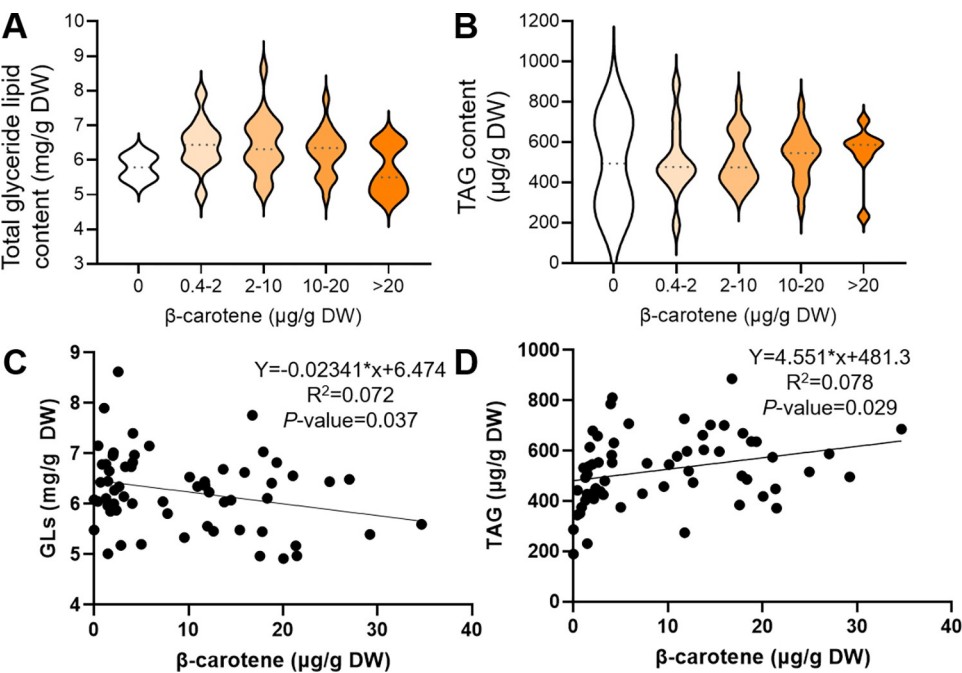

**Fig 5. Quantification of mono-, di- and triglyceride lipids in cassava roots of population GM3732/GM3736 harvested in 2020.** The content of the 17 most intense glyceride lipids detected (A) and six most intense TAGs detected (B) was displayed as a violin chart. The cassava varieties were grouped based on their β-carotene content in five groups (see x-axis). The data was also displayed as a XY graph to show the correlation to β-carotene (C, D). Results from linear regression were displayed as Chi squared goodness of fit (R²) and significance (*P*-value).

and linear regression was performed to confirm this result. Total glyceride lipid content had a negative Pearson coefficient (-0.268) and TAG content a positive Pearson coefficient (0.28). Both correlation analyses were statistically significant with *P*-values 0.037 and 0.029, respectively, and had a coefficient of determination ($R^2$) 0.072 and 0.078, respectively (Fig 5C and 5D).

For GC-MS analysis, cassava roots, harvested in 2017 and 2020 for subcellular fractionation, were analysed. The polar extracts contained 46 identified primary metabolites, including mono- and di-saccharides, amino acids, intermediary metabolites of the TCA cycle and organic acids; and 17 Level 3 and 4 unidentified metabolites. Principal component analysis (PCA) was performed for both harvest years individually (Fig 6A and 6B). Both harvests showed similar cluster patterns of the varieties and indicated underlying chemotypes based on their β-carotene content. The loading plots indicated several metabolites influencing the cluster of high (GM3736-29, GM3736-78 and GM905-57), medium (GM3736-44, GM3736-71 and GM905-60) and low β-carotene varieties (GM3736-2, GM3736-50). Isoleucine, threonine, valine, phenylalanine, glutamic acid, myristic acid, stearic acid, and monostearin were associated with high β-carotene content. Citrulline, alanine and gluconic acid were associated with medium β-carotene content and mannose, ribose and erythronic/threonic acid with low β-carotene content. Correlation analysis was performed and only three metabolites showed a statistically significant Pearson coefficient to β-carotene for both harvests (2017 and 2020): stearic acid (0.6 and 0.9), gluconic acid (-0.6 and -0.8) and erythronic/threonic acid (-0.6 and -0.7).

Metabolite profiling by LC-MS is a common technique to analyse polar metabolism and was applied in the present study to elucidate potential chemotypic differences between cassava varieties with different amounts of β-carotene (Fig 7). Three different cassava population bred for increased β-carotene content were used and included the GM3732/3736 population

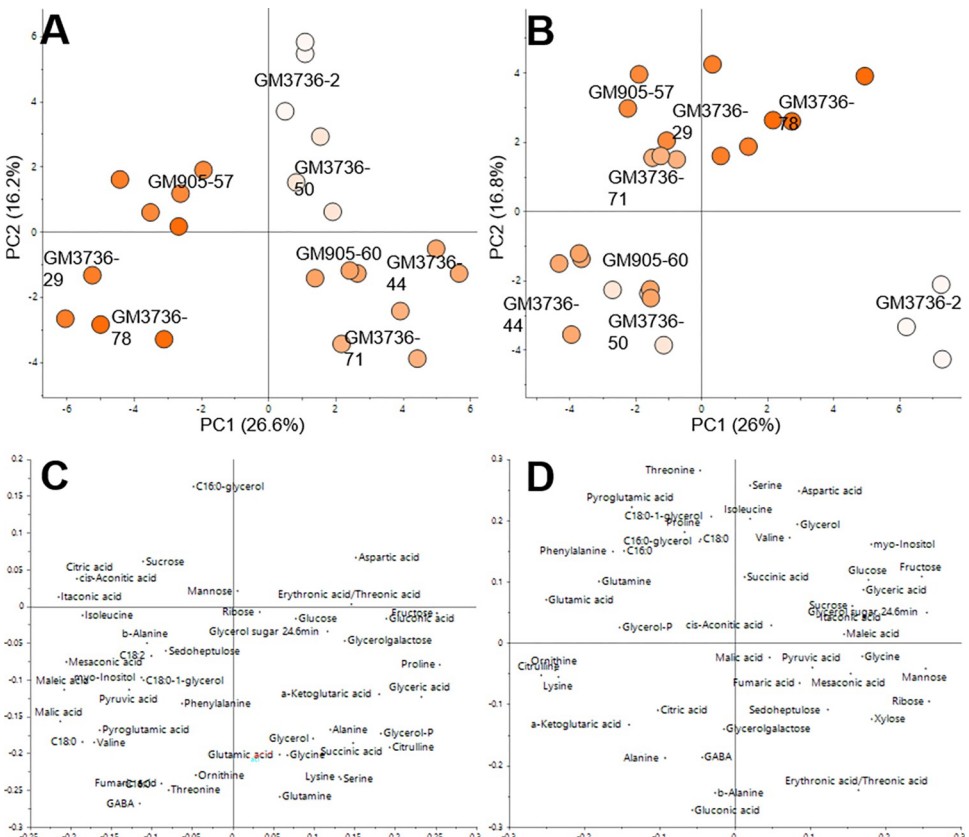

**Fig 6.** GC-MS analysis of cassava roots with eight different root phenotypes cultivated over two years (2017 (A) and 2020 (B)). The β-carotene content of each variety is indicated by a colour range of white (lowest) to orange (highest). The colour range was scaled to the lowest and highest β-carotene of each crop: 0.6–28 μg/g DW (A) and 0–34.7 μg/g DW (B). Three biological replicates were analysed for each variety and are displayed individually. The loading plots are displayed below the respective score plots (2017 (C), 2020(D)).

harvested in 2020 and used for TAG analysis and two populations derived from similar parents (GM5270 and GM5309). LC-MS analysis detected ~10000 molecular features in the polar cassava root extracts and showed very similar results for the three cassava populations. PCA score plots were coloured by β-carotene content and showed no metabolite profile based on β-

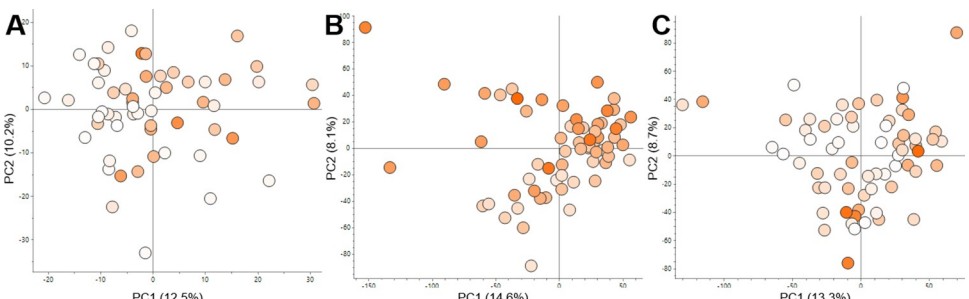

**Fig 7. Metabolite profiling of three carotenoid breeding populations.** The β-carotene content of each variety is indicated by a colour range of white (lowest) to orange (highest). The colour range was scaled to the lowest and highest β-carotene of each crop: (A) GM3732/GM3736 with 0–34.7 μg/g DW, (B) GM5309 with 3.4–13.7 μg/g DW and (C) GM5270 with 0–21.6 μg/g DW. Varieties are displayed as an average of three technical replicates.

carotene content. The first two principal components (PCs) represented ~22% of the molecular features for all three populations. Score plots with PC1 and the other seven to nine PCs showed the same lack of differentiation based on β-carotene content. The proximate analysis indicated a corelation of starch and β-carotene content in eight varieties harvested in 2020. Therefore, glucose and sucrose amounts were established from the metabolite profiling for correlation analysis to β-carotene levels (S2 Fig). For populations GM5309 and GM5270, dry matter content was measured and also correlated to β-carotene. Simple linear regression for glucose, sucrose and dry matter established either a low, statistically significant goodness-of-fit ($R^2 < 0.14$) or a high, not statistically significant goodness-of-fit ($R^2 > 0.5$) to β-carotene.

## Discussion

The present study has characterised a cassava breeding population designed to deliver donor accessions with enhanced provitamin A (β-carotene) content. The data showed the predominance of β-carotene over α-carotene in cassava root tissue. These data suggested the β-branch of the carotenoid pathway was biosynthetically favoured. The correlation of phytoene with β-carotene content could also provide further evidence that phytoene synthase is an influential biosynthetic step in the pathway, corroborating previous findings [19, 20, 40]. In contrast to the relationship between phytoene and β-carotene, apocarotenoids were present in all the selected lines with altered carotenoid content, but no correlation was evident between anabolic carotenoids and their catabolic products (e.g., apocarotenoids) in cassava roots. These data would suggest complex regulatory processes are operational beyond precursor/product interactions that are often attributed to transcriptional regulation. The independence of apocarotenoid formation could relate to carotenoid derived phytohormone biosynthesis and other essential signalling processes [12, 41–43]. These present findings are contrary to previous studies focused on a small number of cassava genotypes, whereby total apocarotenoid content or particularly phytohormones (e.g. abscisic acid) exhibited a direct correlation between β-carotene content and derived metabolites [17, 43]. More recent studies of cassava carotenoid breeding populations reported a highly variable carotenoid content in root colour phenotypes [14, 15, 44]. These studies established a high heritability of carotenoid content based on carotenoid genes (e.g. accumulation process) and non-carotenoid genes (e.g. precursor transport to roots) and epistatic effects. All contributing to the high variability of carotenoid content and lack of correlation between carotenoid biosynthesis and degradation.

Proteomic analysis of different cassava root phenotypes indicated that the majority of metabolic processes in the isolated plastids were amyloplast related, such as starch biosynthesis, carbohydrate metabolism and energy production [45]. These findings indicated that the technique was robust with respect to amyloplast enrichment. The presence of carotenoid biosynthetic related proteins in the amyloplasts was not observed. This could be a consequence of the lack of available resources, for example the UniProt database for *Manihot esculentum* (Taxon ID 3983) contains only basic annotation for ~98% of the >38000 protein entries. Another reason for the lack of identification could be that the protein quantity in the subcellular fractions was below the level of detection for the analytical method used.

Overall, the profile of the sucrose gradients of cassava amyloplast components was more akin to potato than sweet potato, whereby the later can store ≥400 μg/g DW β-carotene in the amyloplasts [21, 22]. Through the complementation of the proteomic data with lipidomic profiling across the gradient, the coloured band in the middle of the sucrose gradient was confirmed as a membranous structure and the bottom fractions represented the stroma [21, 31, 45]. The top layer of the gradient could not be identified as plastoglobules through the usual marker proteins plastoglobulin and/or fibrillin [22, 31, 46, 47]. However, the uncharacterised

protein present in the top fractions (1 and 2) could be a yet to be identified plastoglobule associated protein. Alternatively, it could be that the amyloplasts found in cassava do not contain or have the capacity to produce plastoglobuli like structures. Instead, alternative macromolecular structures could be formed.

The lipid data were used to categorise the membrane structures separated through the sucrose gradient. Fractions 20–23 could potentially represent the bilayered envelope membranes, as the fractions contained higher levels of DGDG and phospholipids [48]. In particular, lyso-PC, PE and PI were present in higher percentages in fractions 22 and 23, which suggests a separation of the traditional inner and outer envelope membranes in the coloured band of the sucrose gradient [49, 50]. Fractions 1 and 2 contained a higher percentage of MGDG compared to fractions 20–23, which could indicate a monolayer lipid structure such as that attributed to traditional plastoglobuli structures [51]. The two top fractions also had the highest percentage of SQDG and similar levels of PC compared to fractions 20–23. The anionic SQDG determines the surface charge and mediates lipid-protein interactions and membrane fusion, whereas PC is a precursor for galactosyl lipids [52]. Both lipids have a cylindrical shape and tend to form bilayered structures [48]. Traditionally, plastoglobules comprise a neutral lipid centre surrounded by a lipid monolayer containing selected proteins such as plastoglobulin [51, 53, 54]. The high MGDG content could provide more flexibility of the droplet membrane to accommodate the presence of carotenoids. The different lipid composition and lack of plastoglobulin like proteins associated with the lipid droplets could also cause a threshold of β-carotene sequestration in cassava roots at ≤100µg/g DW. Genetic engineering approaches and/or comparisons to starchy crops containing plastoglobulin like proteins (e.g. potato and sweet potato) could aid in providing data to validate this hypothesis.

The three genotypes (GM905-57, GM3736-29 and GM3736-78) with the highest β-carotene content (>10 µg/g DW) had a higher percentage (increase of 3–6%) of PG and PA in the envelope-like membrane fractions. These two phospholipids are precursors for galactolipids, which in combination with increased MGDG levels showed a direct correlation between β-carotene and lipid content in the amyloplast envelope like membrane [49]. An increase of lipid content was observed for amyloplasts as well as the whole tissue of cassava roots and corroborates the hypothesis that more lipids are required to store β-carotene [31, 55]. Another lipid with a direct correlation to β-carotene in the amyloplast fractions was PC. PC is produced in the ER and used as a precursor for MGDG and the sole source of some fatty acids for TAG biosynthesis in the plastid [50, 56–59]. This indicates the increase of another galactolipid precursor as a result of higher β-carotene levels. However, the present data does not provide adequate evidence to ascertain if the lipid changes are a response to increased β-carotene content or more β-carotene can be stored in the plastid due to higher lipid levels. Recently in tomato it has been shown that altered carotenoid content does effect/alter lipid composition in a quantitative and qualitative manner [60].

The contradiction between the data of eight selected genotypes and the whole breeding population, with regards to carotenoid and apocarotenoid presence as well as lipid composition, is likely to be derived from epistatic effects on β-carotene content in cassava root tissue [15, 44]. The measurement of carotenoids in the same genotypes over several harvest years (S2 Table) also suggests an environmental influence on β-carotene content, which naturally occurs for micronutrients in the same growth cycle [61]. Furthermore, the fluctuation of β-carotene content indicates the capacity of cassava root tissue to store up to ~40 µg β-carotene /g DW without detrimental effects on the metabolism. This was inferred by the untargeted analysis of three different breeding populations, which showed a lack of separation/clustering between low and high β-carotene producers. This was furthermore supported through the lack of correlation of β-carotene to glucose, sucrose, and dry matter content, contrary to previous

observations in three genetically modified cassava genotypes [17]. These transgenic cassava roots produced only up to 60 μg/g DW β-carotene, despite an additional modification for increased expression of the methylerythritol 4-phosphate (MEP) pathway. The modification could have affected the metabolic flux and diverted carbon sources from starch biosynthesis to carotenoid production. Presumably, this has not occurred in the conventional breeding strategy of the carotenoid population analysed in the present study, based on the correlation analysis with dry matter content. The highest β-carotene producer in the present study contains ~16 μg/g fresh weight, which represents 1.3 μg retinol activity equivalents (RAE) and is equivalent to 0.2% of the recommended daily allowance for vitamin A. Therefore, ~500 g of the best cassava lines would need to be integrated into the daily diet for sufficient levels of provitamin A.

## Conclusions

In conclusion, the metabolomic analysis of cassava breeding populations has identified natural variation for β-carotene content, other carotenoids and apocarotenoids. Lines with levels of β-carotene up to 40 μg/g DW were identified. In addition, these lines did not have reduced dry matter, but showed changes in the total lipid content and lipid composition of amyloplast membranes. Collectively, the proteomic and metabolomic data indicated that cassava roots are predisposed to starch formation and accumulation. Thus, extensive breeding and/or engineering of biology will be required to development cassava lines as highly impactful biofortified sources of provitamin A in its own right. However, in combination with other diverse dietary components, the biofortified cassava lines will contribute to the eradication of vitamin A deficiency. Future studies are required to elucidate the proposed intrinsic mechanism for carotenoid production in cassava amyloplasts and should consider the potential contradiction of selected and whole population panels.

## Supporting information

**S1 Fig. Chromatograms of lipid analysis of all fractions of GM3736-71 and GM3736-78.**
(PDF)

**S2 Fig.** Correlation analysis of β-carotene to glucose, sucrose and dry matter of breeding populations (A) GM3732/3736 in 2020, (B) GM5309 and (C) GM5270.
(PDF)

**S1 Table. Parental background for cassava carotenoid populations in the present study.**
Breeding strategy for GM3732 and GM3736 is displayed as an ancestral tree and contribution of parental lines to GM5270 and GM5309 is listed as percentage.
(DOCX)

**S2 Table. Carotenoid content of all cassava lines analysed in the present study (μg/g DW).**
(XLSX)

**S3 Table. Data from GC-MS analysis of GM3732/3736 cultivated in 2020 (area relative to internal standard and sample weight).**
(XLSX)

**S4 Table. LC-MS data of GM5309 lines (area relative to internal standard and sample weight).**
(XLSX)

**S5 Table. LC-MS data of GM5270 (area relative to internal standard and sample weight).**
(XLSX)

**S6 Table. LC-MS data of GM3732/3736 cultivated in 2020 (area relative to internal standard and sample weight).**
(XLSX)

**S7 Table. Data of identified compounds from SPME-GC-MS analysis (area relative to internal standard).**
(XLSX)

**S8 Table. Data from glyceride lipid analysis of GM3732/3736 cultivated in 2020 (areas and mg/g DW of total content).**
(XLSX)

**S9 Table. Data from lipid analysis of sucrose gradient fractions for eight select lines (mg/g DW).**
(XLSX)

**S10 Table. Pearson correlation analysis of lipid content from sucrose gradient fractions to β-carotene content for eight selected lines.** Data includes correlation values, *P*-values and coefficient of determination. Significant correlation values are displayed in bold and coefficients >0.5 were underlined.
(XLSX)

**S11 Table. List of proteins identified in sucrose gradient of amyloplasts of GM905-60 (M), GM3736-71 (L), GM3736-78 (H).**
(PDF)

## Acknowledgments

We would like to thank C. Gerrish for excellent technical support; Drs E.M.A. Enfissi, L. Perez-Fons, M. Nogueira and H.M. Berry for their advice and support during this study and all donors who supported this research through their contributions to the CGIAR Fund (https://www.cgiar.org/funders/).

## Author Contributions

**Conceptualization:** Margit Drapal, Luis Augusto Becerra Lopez-Lavalle, Paul D. Fraser.

**Data curation:** Margit Drapal.

**Formal analysis:** Margit Drapal.

**Funding acquisition:** Luis Augusto Becerra Lopez-Lavalle, Paul D. Fraser.

**Investigation:** Margit Drapal, Tatiana M. Ovalle Rivera.

**Methodology:** Margit Drapal.

**Project administration:** Luis Augusto Becerra Lopez-Lavalle, Paul D. Fraser.

**Resources:** Tatiana M. Ovalle Rivera, Luis Augusto Becerra Lopez-Lavalle, Paul D. Fraser.

**Supervision:** Luis Augusto Becerra Lopez-Lavalle, Paul D. Fraser.

**Validation:** Margit Drapal, Tatiana M. Ovalle Rivera.

**Visualization:** Margit Drapal.

**Writing – original draft:** Margit Drapal.

**Writing – review & editing:** Margit Drapal, Luis Augusto Becerra Lopez-Lavalle, Paul D. Fraser.

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
