## [Decision Letter · Decision Letter 0]

20 Sep 2024

PONE-D-24-28459

Carotenoid composition and sequestration in cassava ( Manihot esculentum Crantz) roots

PLOS ONE

Dear Dr. Drapal,

Thank you for submitting your manuscript to PLOS ONE. After careful consideration, we feel that it has merit but does not fully meet PLOS ONE’s publication criteria as it currently stands. Therefore, we invite you to submit a revised version of the manuscript that addresses the points raised during the review process.

We look forward to receiving your revised manuscript.

Kind regards,

Shashi Kant Bhatia

Academic Editor

PLOS ONE

Journal Requirements:

Reviewers' comments:

Reviewer's Responses to Questions

**Comments to the Author**

1. Is the manuscript technically sound, and do the data support the conclusions?

Reviewer #1: Yes

Reviewer #2: No

Reviewer #3: Yes

Reviewer #4: Yes

Reviewer #5: Yes

2. Has the statistical analysis been performed appropriately and rigorously? 

Reviewer #1: Yes

Reviewer #2: No

Reviewer #3: Yes

Reviewer #4: Yes

Reviewer #5: Yes

3. Have the authors made all data underlying the findings in their manuscript fully available?

Reviewer #1: Yes

Reviewer #2: No

Reviewer #3: Yes

Reviewer #4: Yes

Reviewer #5: Yes

4. Is the manuscript presented in an intelligible fashion and written in standard English?

Reviewer #1: Yes

Reviewer #2: No

Reviewer #3: Yes

Reviewer #4: Yes

Reviewer #5: Yes

5. Review Comments to the Author

Reviewer #1: I went through the manuscript and the data provided. I found the study to be in-depth and the details were meticulously presented by the authors in the paper. A thorough study has been done and the paper is suitable for acceptance for publication.

Reviewer #2: The author discusses the relationship between the amount of β-carotene in cassava roots and β-carotene’s precursors, as well as lipids, starch, and proteins, using several cassava breeding lines. However, because β-carotene is synthesized via the mevalonate pathway, the explanation regarding its relationship with lipids is lacking in detail. Metabolites were quantified using MS, and PCA analysis was conducted, but please mention what was revealed through the PCA analysis. Additionally, correlation analysis was performed, so please mention what was discovered through these analyses regarding to the connection on metabolic pathway between β-carotene and its precursors, as well as other metabolites. A concise conclusion is needed for each subsection in the Results. It would be helpful to narrow the focus of the discussion further. For example, in Line 68-69 of the introduction: "However, the characterization of amyloplasts, the site of starch and carotenoid biosynthesis and sequestration in cassava roots, has not been carried out." Please refer to this point more specifically. Furthermore, regarding the quality of the data, for instance, the starch content is only 100-200 mg per 1 g of dry weight (DW). There are serious concerns about the quality of the data.

Reviewer #3: The study presented in the manuscript is of high quality and makes a valuable contribution to the field. The authors have conducted a thorough investigation and have addressed the research question with appropriate methodology and analysis. The data analysis is sound, and the findings are well-supported by the provided evidence.

While I have identified a few minor suggestions for improvement, they do not detract from the overall quality and impact of the study.

Comments to the authors:

1- The two references listed in line 65 (e.g. Jaramillo et al.,66 2022; Welsch et al., 2010) weren’t numbered (19 and 20 in the order of the references…. Please check and add their numbers.

2- Additionally, in line 429 (Arango et al., 2010; Jaramillo et al., 2022; 430 Welsch et al., 2010) list of references must be numbered.

3- The data listed in Table 1 is the protein name in the first raw 14_3_3 domain……… or 14-3-3 domain…. Please check.

Reviewer #4: Dear authors,

this work is interesting the quality of figures is good and can be published in this journal.

best regard

Reviewer #5: I went through the manuscript and found it to be interesting for publication. The study was conducted following a proper experimental setup, and the results were presented neatly. I would like to congratulate the authors for doing such wonderful research and presenting it in such a clear way.

6. PLOS authors have the option to publish the peer review history of their article (what does this mean?). If published, this will include your full peer review and any attached files.

Reviewer #1: No

Reviewer #2: No

Reviewer #3: **Yes: **Fatma Mohamed Abd El-Mordy

Reviewer #4: **Yes: **Sakineh Abbasi

Reviewer #5: No

---

## [Author Response · Author response to Decision Letter 0]

7 Oct 2024

Review Comments to the Author

Response: We would like the reviewers for their positive feed-back and acknowledgement of the work performed. As per Editor’s request, the manuscript was formatted to the guidelines of PLOS ONE and the funders information removed from the Acknowledgement section.

The response to reviewers include references to the text of the “clean” manuscript.

Reviewer #2: The author discusses the relationship between the amount of β-carotene in cassava roots and β-carotene’s precursors, as well as lipids, starch, and proteins, using several cassava breeding lines. However, because β-carotene is synthesized via the mevalonate pathway, the explanation regarding its relationship with lipids is lacking in detail. Metabolites were quantified using MS, and PCA analysis was conducted, but please mention what was revealed through the PCA analysis. Additionally, correlation analysis was performed, so please mention what was discovered through these analyses regarding to the connection on metabolic pathway between β-carotene and its precursors, as well as other metabolites. A concise conclusion is needed for each subsection in the Results. It would be helpful to narrow the focus of the discussion further. For example, in Line 68-69 of the introduction: "However, the characterization of amyloplasts, the site of starch and carotenoid biosynthesis and sequestration in cassava roots, has not been carried out." Please refer to this point more specifically. Furthermore, regarding the quality of the data, for instance, the starch content is only 100-200 mg per 1 g of dry weight (DW). There are serious concerns about the quality of the data.

Response: It is surprising that the reviewer seems to be confused with isoprenoid biosynthesis in the context of the present manuscript. The majority of precursors (IPP and DMAPP) for beta-carotene are produced exclusively from the 2-C-methyl-D-erythritol 4-phosphate (MEP) pathway in the plastid. The cytosolic mevalonate (MVA) pathway forms IPP separately to serve non-plastid derived isoprenoids (Fraser and Bramley 2004;doi:10.1016/j.plipres.2003.10.002). The TCA cycle is a central metabolic pathway feeding into carotenoid, lipid and starch biosynthesis, via common intermediary metabolites. Hence, it is speculated in Line 505 that the significantly lower starch content of transgenic cassava in Beyene et al 2018 (Reference 17) could be the result of a metabolic remodelling. However, this was not the case in the present study (Line 506).

The relationship between beta-carotene and lipid content has been explained in Line 487-490. Currently, the scientific community has not yet elucidated whether changes in carotenoid levels lead to an increase in lipid content or an increased lipid content allows for more carotenoids to be stored. Hence, the discussion describes the positive correlation of beta-carotene and lipids observed and lists a recent study in tomato with similar results.

The results of the PCA and correlation analysis were listed in Lines 238-242; 392-403; 410-419. The discussion of these results indicates a complex regulatory process of carotenoids and their precursors and products (Line 426-440). The lack of detrimental effects of higher beta-carotene levels on the starch and sugar/carbohydrate content was discussed in Line 497-503. 

The point of “characterisation of amyloplasts has not been carried out” was addressed specifically and multiple times with regards to (i) structure is similar to potato Line 450-452 and contains two lipid layers Line 452-459 and (ii) the majority of metabolic processes in the amyloplast are starch related Line 441-443.

The results from the present study are very complex. Therefore, the discussion was focused on precursors for carotenoid production, followed by the characterisation of the plastid and potential links of beta-carotene levels to lipid and starch content in a non-transgenic, heterogenic breeding population for beta-carotene.

The data presented was based on established methods previously published by the authors and its quality previously reviewed multiple times in the peer-review process. The reviewer’s concerns for the low starch content have merit. However, the data from multiple crops show similar levels and are comparable to previously published cassava data (variety IAC 06-01 in Schmitz et al 2016, https://doi.org/10.1002/star.201600272). This could be a result of the breeding strategy for this particular population.

Reviewer #3: The study presented in the manuscript is of high quality and makes a valuable contribution to the field. The authors have conducted a thorough investigation and have addressed the research question with appropriate methodology and analysis. The data analysis is sound, and the findings are well-supported by the provided evidence.

While I have identified a few minor suggestions for improvement, they do not detract from the overall quality and impact of the study.

Comments to the authors:

1- The two references listed in line 65 (e.g. Jaramillo et al.,66 2022; Welsch et al., 2010) weren’t numbered (19 and 20 in the order of the references…. Please check and add their numbers.

2- Additionally, in line 429 (Arango et al., 2010; Jaramillo et al., 2022; 430 Welsch et al., 2010) list of references must be numbered.

3- The data listed in Table 1 is the protein name in the first raw 14_3_3 domain……… or 14-3-3 domain…. Please check.

Response: We thank the reviewer for highlighting the formatting error. The references were formatted to the guidelines of PLOS ONE. The protein name in the table was changed to 14-3-3 domain in Table 1.

---

## [Editor Report · Decision Letter 1]

9 Oct 2024

Carotenoid composition and sequestration in cassava ( Manihot esculentum Crantz) roots

PONE-D-24-28459R1

Dear Dr. Fraser,

We’re pleased to inform you that your manuscript has been judged scientifically suitable for publication and will be formally accepted for publication once it meets all outstanding technical requirements.

Kind regards,

Shashi Kant Bhatia

Academic Editor

PLOS ONE

---

## [Editor Report · Acceptance letter]

6 Nov 2024

PONE-D-24-28459R1 

PLOS ONE

Dear Dr. Fraser, 

I'm pleased to inform you that your manuscript has been deemed suitable for publication in PLOS ONE. Congratulations! Your manuscript is now being handed over to our production team.

Kind regards, 

on behalf of

Dr. Shashi Kant Bhatia 

Academic Editor

PLOS ONE